# Neuromuscular Fatigue Responses of Endurance- and Strength-Trained Athletes during Incremental Cycling Exercise

**DOI:** 10.3390/ijerph19148839

**Published:** 2022-07-21

**Authors:** Maciej Jurasz, Michał Boraczyński, Zbigniew Wójcik, Piotr Gronek

**Affiliations:** 1Department of Sport Medicine and Traumatology, Poznan University of Physical Education, 61-871 Poznań, Poland; maciej@archeus.pl; 2Faculty of Health Sciences, Collegium Medicum, University of Warmia and Mazury in Olsztyn, 10-719 Olsztyn, Poland; 3Faculty of Health Sciences, Olsztyn University, 10-283 Olsztyn, Poland; zbigniew-wojcik@o2.pl; 4Laboratory of Healthy Aging, Department of Dance and Gymnastics, Poznan University of Physical Education, 61-871 Poznań, Poland; gronek@awf.poznan.pl

**Keywords:** neuromuscular fatigue, muscle bioelectrical activity, rating of perceived exertion, cycling exercise

## Abstract

This study explored the development of neuromuscular fatigue responses during progressive cycling exercise. The sample comprised 32 participants aged 22.0 ± 0.54 years who were assigned into three groups: endurance-trained group (END, triathletes, *n* = 10), strength-trained group (STR, bodybuilders, *n* = 10) and control group (CG, recreationally active students, *n* = 12). The incremental cycling exercise was performed using a progressive protocol starting with a 3 min resting measurement and then 50 W workload with subsequent constant increments of 50 W every 3 min until 200 W. Surface electromyography (SEMG) of rectus femoris muscles was recorded during the final 30 s of each of the four workloads. During the final 15 s of each workload, participants rated their overall perception of effort using the 20-point rating of the perceived exertion (RPE) scale. Post hoc Tukey’s HSD testing showed significant differences between the END and STR groups in median frequency and mean power frequency across all workloads (*p* < 0.001 and *p* < 0.01, respectively). Athletes from the END group had significantly lower electromyogram amplitude responses than those from the STR (*p* = 0.0093) and CG groups (*p* = 0.0006). Increasing RPE points from 50 to 200 W were significantly higher in the STR than in the END group (*p* < 0.001). In conclusion, there is a significant variation in the neuromuscular fatigue profiles between athletes with different training backgrounds when a cycling exercise is applied. The approximately linear trends of the SEMG and RPE values of both groups of athletes with increasing workload support the increased skeletal muscle recruitment with perceived exertion or fatiguing effect.

## 1. Introduction

Scientific interest in muscle stimulation (myoelectric activity) and fatigue in athlete performance has increased in recent years [1]. Neuromuscular fatigue can occur via central and/or peripheral sites along the motor pathway of force production. In general, central or neural fatigue (affecting the proximal motor neurons of the brain and spinal cord) involves insufficient neural drive to the muscles, whereas peripheral fatigue, where the motor units (MUs) of the peripheral nerves, motor endplates and muscle fibers are affected, involves changes beyond the neuromuscular junction [2]. Several previous cross-sectional studies have reported that specific power, strength and endurance modes of training influence respective neuromuscular fatigue (NMF) profiles in trained athletes [2,3]. Moreover, within endurance-trained athletes, the type of locomotion may vary with the specific sport and further influence the NMF profile [2]. There are a number of mechanical (e.g., decreased countermovement jump height) and metabolic (e.g., lactate accumulation) measures of fatigue [4,5]. In addition, psychophysiological variables such as the rating of perceived exertion (RPE),which is related to physiological markers of the stress response to exercise, are widely used. RPE has been used to investigate the mechanisms of fatigue as well as to prescribe exercise protocols in numerous populations [6,7,8]. Another common, non-invasive and objective method to measure neuromuscular fatigue is electromyography (EMG). While EMG can be assessed both via surface electrodes and intramuscularly, for the purposes of this paper we will focus on conventional bipolar surface EMG (SEMG). In general, SEMG provides a global view of skeletal muscle function as well as the activity of individual MUs [9]. The analysis of multielectrode recordings via SEMG sensors, which display muscle activation patterns (the algebraic summation of all MU action potentials), has been extensively used in the fields of sports science and medical research.

Of interest is that the RPE responses during resistance and endurance performance have been associated with the degree of skeletal muscle recruitment measured by SEMG [10]. Multiple studies have shown a linear increase in the SEMG and RPE during exhaustive constant- and progressive-load exercise [11,12], which allows the estimation of their increasing rates and helps to estimate the effect of engaged muscle mass on the magnitude of performance fatigability. Increased SEMG activity is a good indicator of a change in muscular neural drive [13] and has been used to interpret both dysfunctional and functional muscle recruitment patterns related to cycling activity [14]. For high-intensity cycling exercise, a relatively consistent magnitude of peripheral fatigue has been shown in various exercise protocols [15]. However, this consistency of peripheral fatigue (especially at exercise termination), which reflects a centrally mediated “individual critical threshold”, is probably task- or training-dependent [16,17]. To the best of our knowledge, NMF profiles and specific adaptive differences between cardiovascular endurance and muscular strength modes of training have not been adequately investigated in moderately experienced athletes. Hence, the question arises whether differences in muscle activity, subjective fatigue perception and associated decrements in cycling performance exist during a submaximal incremental cycling test in athletes trained in triathlon and bodybuilding, and recreationally active men. Because changes in muscle activation patterns and kinematics under dynamic fatigue have been previously observed, identifying any specific training-dependent differences would be beneficial for determining the likelihood of musculoskeletal injury and for designing appropriate motor control training protocols.

This study was designed to investigate and compare the development of exercise-induced neuromuscular and perceptual fatigue responses using SEMG and RPE measurements during an incremental submaximal cycling exercise. We hypothesized that the specific training background (cardiovascular endurance versus muscular strength) would influence the neuromuscular fatigue profiles, as expressed by SEMG activity and RPE-based subjective fatigue perception in endurance- and strength-trained athletes and recreationally active men.

## 2. Materials and Methods

### 2.1. Participants

A total of forty-seven male participants (mean ± SD: age 22.0 ± 0.5 years, range 19–25 years) from Poznań in the western region of Poland were recruited to take part in a cross-sectional observational study. The recruitment process was conducted by academic website information, email and recruitment flyers. The thirty-two eligible participants were placed into their three respective groups based on their training history. The endurance-trained group (END; triathletes, *n* = 10, age 20.3 ± 0.6 years, body mass 75.1 ± 2.9 kg, body height 179.8 ± 2.1 cm, body mass index 23.2 ± 0.64 kg∙m^–2^) reported training regularly for 3–7 years. Their weekly training volumes averaged 14.5 ± 1.3 h, with 1–2 daily exercise sessions in the three triathlon disciplines (4.9 ± 0.3 h for swimming, 5.7 ± 0.6 h for cycling, 3.1 ± 0.3 h for running and 0.8 ± 0.1 h for specific preparation). Their average training intensity was 15–17 on the RPE scale.

The strength-trained athletes (STR; bodybuilders, *n*= 10, age 22.4 ± 0.4 years, body mass 83.9 ± 2.7 kg, body height 181.9 ± 2.4 cm, body mass index 25.4 ± 0.62 kg∙m^–2^) reported training 5.1 ± 0.2 sessions/week (2.4 ± 0.3 h/session) with 4–7 years experience in consistent bodybuilding exercise (minimum of three sessions per week lifting workouts). Their training intensity rating was 15–16 on the RPE scale. Typically, training programs reported by the participants in STR group consisted of multi-joint weightlifting exercises to promote sufficient stimuli for gaining muscle size and strength in all muscles involved in the exercise. The bodybuilders performed 2–3 circuits of 10–12 exercises (total-, upper-, and lower-body muscles), with each set consisting of 6–12 repetitions.

The recreationally active control individuals (CG, *n* = 12, age 23.2 ± 0.4 years, body mass 78.2 ± 2.6 kg, body height 181.6 ± 1.8 cm, body mass index 23.7 ± 0.64 kg∙m^–2^) consisted of students participating in a variety of recreational activities. Participants from CG group reported the frequency and duration of different types of activity: vigorous (i.e., heavy lifting, performing high-intensity exercises, using treadmill); moderate (i.e., carrying light loads and bicycling at a regular pace); walking activities and the average time spent sitting on a weekday. According to the IPAQ scoring recommendations, these participants were classified as moderately active (≥600 MET–minutes/week). Their mean total MET score was 954 ± 287.3 MET–minutes/week.

### 2.2. Eligibility Criteria

The inclusion criteria for END and STR groups were: male, aged 18–30 years, self-reported regular engagement in endurance or strength training, at least 3 years of competitive experience, no current sustained experience in the other modes of training, non-smokers, not taking performance-enhancing drugs and reporting no medical contraindications to participate in exercise testing (e.g., knee injury or neuromuscular disorder). For the CG group the inclusion criteria included: male, aged 18–30 years, non-smokers, no reported medical contraindications to participate in exercise testing, a BMI < 25 kg/m^2^ and active lifestyle (>600 MET) confirmed by an adapted version of short-form of International Physical Activity Questionnaire (IPAQ-SF). Participants from CG group were defined as recreationally active if they performed regular exercise (moderate-to-vigorous exercise 2–3 times per week), but did not train or compete within a sport at a competitive level.

### 2.3. Ethical Approval

All participants gave written informed consent and verbal assent regarding their participation in the study. The research and protocol were approved by the Scientific Ethical Committee of the Karol Marcinkowski University of Medical Sciences in Poznań. The procedures performed in this study involving human participants were in accordance with the ethical guidelines of the 2013 World Medical Association Declaration of Helsinki (2013: seventh revision, 64th meeting, Fortaleza, Brazil).

### 2.4. Design and Procedures

Data collection was performed over a 2-week period, in which participants took part in laboratory tests on two separate occasions with a minimum of 48 h between sessions. During the first visit to the laboratory, anthropometric data were collected, and all participants underwent a non-invasive pre-participation cardiovascular screening (based on 12-lead ECG). During the second visit, participants had a familiarization session (cycle ergometer and the testing procedure) and performed incremental cycling test. Testing was conducted in the morning hours (9:00–11:00). Participants performed tests in condition-controlled laboratory room with a fan (average temperature: 22.5 ± 0.5 °C, relative humidity: 35.0 ± 5.0%, barometric pressure: 760–770 mmHg).

### 2.5. Pre-Test Preparation

Participants were required to refrain from strenuous exercise during the 48 h prior to the incremental cycling test session. No alcohol, caffeine and tobacco were allowed for 24 h prior to testing. In addition, participants were asked to avoid using any ergogenic aids for at least 48 h before the testing session. They were also instructed to refrain from eating for 2-h prior to testing and only drink water.

### 2.6. Anthropometry

Body height (BH) was measured to the nearest 0.1 cm using a patient weighing scale with a height rod (Seca 217, Hamburg, Germany). Body mass (BM, after removal of shoes and heavy clothing) was measured to the nearest 0.1 kg.

### 2.7. Incremental Cycling Test

The exercise test was performed using the electromagnetically braked cycle ergometer (Ergometrics 900 S, Ergo-line GmbH, Bitz, Germany). Before the study, the cycle ergometer was calibrated for power outputs of 25–1000 W at different cadences and was found to be within 1% of a true value. The cycle ergometer was set according to each participant’s anatomy: the participant adjusted saddle height and the handlebars to his own cycling posture. However, the trunk forward inclination angle did not exceed 20°, and the knee flexion angle was 5° with the extended lower limb. Three-minute standardized warm-up protocol was followed prior to testing. The latter was performed at a constant workload (1.5–2.0 W∙kg^–1^ of BM) and pedaling cadence (70 rotations∙min^−1^) interspersed with 2 all-out sprints of 2–3 s (~90 rotations∙min^−1^) to elicit HR between 150 and 160 beats∙min^−1^. Participants then rested for 15 min prior to incremental cycling test. This graded test was performed using a progressive protocol according to Dufour et al. [18]. The test started with a 3min resting measurement in the sitting position and then 50 W workload with 3 subsequent increments of 50 W every 3 min (100, 150 and 200 W). The cadence was set at 70 rotations∙min^−1^ and was maintained via the use of a metronome.

### 2.8. Ratings of Perceived Exertion (RPE)

During the familiarization session participants were given standard instructions and anchoring procedures for overall perception of effort using the 6–20-point RPE scale [19]. During the final 15 s of each workload, participants were asked to rate their overall perception of effort using any number on a large RPE scale displayed in front of them. A rating of 6 was associated with “no exertion at all” (rest) and a rating of 20 was considered to be a “maximal exertion” of effort and associated with the most stressful exercise ever performed. RPE is widely used in exercise tests using cycle ergometers (incremental aerobic exercise).

### 2.9. Surface Electromyography (SEMG)

Bipolar SEMG is a well-established method for evaluating muscular load. During the laboratory visit, pre-gelled surface electrodes (Ag/AgCl, AccuSensor, Lynn Medical, Wixom, MI, USA) were placed in a bipolar arrangement (20-mm center-to-center) on the dominant rectus femoris (RF) muscles (27 participants were right-legged and 5 left-legged). Dominant limb was determined using methods consistent with those described by Hebbal and Mysorekar [20]. The recording electrodes were fixed lengthwise over the anatomic reference points of the muscle according to the Surface Electromyography for a Non-invasive Assessment of Muscles (SENIAM) recommendations [21]. In detail, the surface electrodes were placed as follows: distal to the midpoint along the line connecting the superior side of the patella and anterior superior iliac spine. Before placing the Ag/AgCl adhesive electrodes, the skin was shaved, cleaned with ether–alcohol solution and antiseptic cotton, and dried to control a low impedance at the skin–electrode interface (*Z*< 5 kΩ).To ensure the reliability of the SEMG signals, the same researcher positioned the electrodes on all the participants. Throughout the incremental cycling exercise, an electromyographic recording system—the Muscle TesterME3000Pwith MegaWin software version 1.41 (Mega Electronics Ltd., Kuopio, Finland)—was used to display and analyze SEMG signals in the RF muscles. The obtained data were directly stored on a disk of a personal computer for further offline analysis. Electrical activity of muscles (quantified in microvolts) was recorded four times: in the last 30 s of the set workload values. The EMG signals were analogically amplified with gain of 50×, band-pass filtered at 10–500 Hz and sampled through a 12-bit analog-to-digital (A/D) converter with a sampling frequency of 1000 Hz. The EMG signal was calculated for 0.5-s segments.

### 2.10. Statistical Analysis

Statistica 13.3 software package [22] and IBM SPSS Statistics ver. 26.0.0.1 [23] were utilized for statistical analysis. All the results considered were reported as mean ± SD. Kolmogorov–Smirnov normality test revealed that all the data were normally distributed.

An analysis of variance (ANOVA) model for repeated measures was used to assess differences between means. A two-way mixed design was applied to test for main and interaction effects: group (END, STR, CG) was used as a fixed factor, while workload level (50, 100, 150 and 200 W) was used as a repeated factor. The sphericity of variance for repeated measures was tested using Mauchly’s test [24]. In the case of lack of sphericity of variance, the epsilon Greenhouse–Geisser correction was applied. Detailed differences between means were tested post hoc using the Tukey HSD test [25]. A significance level was set at α=0.05. The effect size (ES) for each ANOVA effect was presented as a partial eta squared (η^2^_p_). The interpretation of ES was based on benchmarks established by Cohen [26] where d = 0.01 indicates a small effect, 0.06 a medium effect and 0.14 a large effect.

## 3. Results

A total of 32 electromyograms were recorded for RF muscles of a quadriceps performing dynamic work during concentric contractions (collected during the last 30 s of each 4 min workload). Figure 1, Figure 2, Figure 3 and Figure 4 show the comparisons between the mean values ± standard deviation (mean ± SD) of SEMG variables (MF, MPF, AEMG) and RPE obtained (1) at the subsequent workloads (50, 100, 150 and 200 W) and (2) for different groups: endurance-trained athletes (END), strength-trained athletes (STR) and control group (CG).

### 3.1. Median Frequency (MF)

There was no interaction effect for mean MF; however, we found the effect for the workload level (F_(3,87)_ = 15.61, *p* < 0.001, η^2^_p_ = 0.350, power = 0.999). The level of MF increased significantly at the subsequent workloads from 50 to 150 W in each group (for consecutive workload comparisons; Tukey’s HSD test; MF: *p* < 0.001, *p* < 0.0002, respectively). The value of MF did not differ significantly at higher workloads (HSD Tukey’s test: 150 W vs. 200 W, *p* = 0.483). The ANOVA demonstrated a significant group effect for MF (F_(2,29)_ = 13.92, *p* < 0.001, η^2^_p_ = 0.490, power = 0.996). A significant difference was noted between the END and STR groups (Tukey’s HSD test: *p* < 0.001). However, no significant difference was found between the END and CG groups (*p* = 0.505) (Figure 1).

### 3.2. Mean Power Frequency (MPF)

A significant group × workload interaction effect was observed in the SEMG variable, MPF (F_(6,87)_ = 8.98, *p* < 0.001, η^2^_p_ = 0.243, power = 0.984). The two-way ANOVA demonstrated a main effect for group (F_(2,29)_ = 4.65, *p* < 0.001, η^2^_p_ = 0.382, power = 0.958) showing that significant differences in MPF across all workloads were found between the END and STR groups (Tukey’s HSD test: END vs. STR, *p* = 0.0054) as well as the STR and CG groups (Tukey’s HSD test: STR vs. CG, *p* = 0.0022). Conversely, no significant main effect for workload was observed (*p* = 0.441) (Figure 2).

### 3.3. Electromyogram Amplitude (AEMG)

The significant group × workload interaction effect was observed in AEMG (F_(6,87)_= 11.75, *p* < 0.001, η^2^_p_ = 0.448, power = 0.999). The ANOVA using AEMG showed in a significant main effect for group (F_(2,29)_ = 10.65, *p* < 0.001, η^2^_p_ = 0.423, power = 0.981). Furthermore, a main effect for workload was also observed (F_(3,87)_ = 173.02, *p* < 0.001, η^2^_p_ = 0.856, power = 1). In general, AEMG tended to increase at subsequent workloads in every group. Post hoc analyses demonstrated that the athletes from the END group had significantly lower AEMG responses than those from the STR group (*p* = 0.0093) and CG group (*p* = 0.0006). Interestingly, at the highest workload (200 W) there was no significant difference between the END and STR groups (*p* = 0.805), while AEMG increased substantially in the CG group and was significantly different compared to the END and STR groups (both *p*< 0.001) (Figure 3).

### 3.4. Ratings of Perceived Exertion (RPE)

Significant main effects for RPE were demonstrated for both group (F_(2,29)_ = 116.84, *p* < 0.001, η^2^_p_ = 0.890, power = 1) and for workload (F_(3,87)_ = 116.84, *p* < 0.001, η^2^_p_ = 0.887, power = 1). There were also significant interactions between groups (F_(6,87)_ = 15.32, *p* < 0.001, η^2^_p_ = 0.514, power = 1), indicating that the successively increasing RPE points from 50 to 200 W were significantly higher in the STR than in the END group (Figure 4). As expected, participants from the CG group reported that each workload completed was more exhaustive as compared to the groups of athletes (all *p* < 0.001).

## 4. Discussion

A number of differences were observed between the END and STR groups with regards to their neuromuscular responses as demonstrated by SEMG and RPE. Amongst the significant differences found were a larger increase in MF and MPF for the RF portion of the quadriceps of the END athletes (triathletes) compared with the STR athletes (bodybuilders). In contrast, a more pronounced increase in AMG and RPE were observed in the STR group as compared to the END group. However, MF, MPF and RPE responses were the highest in the CG group (recreationally active participants).

According to our assumptions, the neuromuscular responses were significantly different between the END and STR groups, indicating that both central (neural) and peripheral (muscular) function is associated with the training background. However, various neuromuscular factors, including altered patterns in neural recruitment [27] and the intrinsic and extrinsic influences (e.g., thickness of subcutaneous tissue and/or distribution of MU conduction velocities) on the EMG signal could affect the NMF profiles of the study participants [28,29]. In comparison to isometric muscle contraction, diverse biochemical processes (e.g., depletion of high-energy phosphates including creatine phosphate, accumulation of lactate, hydrogen ion [H^+^] and inorganic phosphate) associated with the muscle fatigue changes cause the interpretation of SEMG signals from dynamic contractions to be much more difficult [30]. Since muscle activity has been shown to be repeatable during incremental cycling exercise [31] and we did not use any method of normalization, it could be assumed that the changes found in muscle activity during each cycling workload were due to the variation in muscle activity caused by fatigue or from the intrinsic and extrinsic influences on the EMG signal.

While the same muscle (e.g., RF) of different types of athletes may exhibit various percentages of MUs, these muscles will probably show a different SEMG activity during progressive cycling exercise. Indeed, the tested groups were significantly different in all SEMG variables as well as RPE. Furthermore, endurance- and strength-trained athletes present a heterogeneous make-up of different fiber types (muscle fiber phenotypic profiles) with task-specific function [32]. It is well recognized that highly-trained endurance athletes possess significantly higher slow-twitch muscle fiber (type I, oxidative) percentages (as high as 90–95%) relative to resistance-trained (60–80%) or untrained individuals (approx. 50%) [33,34]. Since endurance training increases the amount of type I fibers [35], the differences in muscle fiber type distribution between the subjects may be potential factors that influence SEMG responses. In addition, in terms of morphological structure, different muscles within a quadriceps represent diversified muscle fiber phenotypic profiles; the proportion of fast-twitch muscle fibers (type II, glycolytic) is highest in RF (70.5%) compared to vastus lateralis (VL) and vastus medialis (VM) (67.3% and 56.3%, respectively) [36]. Nevertheless, an important practical aspect of this research is that RF muscles appear to be most sensitive to fatigue [37].

The comparative analysis of changes in lower extremity muscle activities during cycling in laboratory settings is limited because of the variable experimental study designs, participant training characteristics and/or specific muscles or muscle groups. The majority of studies recorded the activity patterns in VL, which proved to be highly reproducible, as well as it being considered the most active quadriceps component during cycling exercise [38,39,40]. However, RF (a biarticular muscle) and other muscle groups within the quadriceps i.e., VL and VM (monoarticular muscles), and medial and lateral gastrocnemius (MG and LG), are moderately sensitive in tracking changes in workload when a cycling mode is applied [41]. Although during the incremental cycling exercise the integrated EMG (iEMG) signal was consistently lower in RF compared with VL and VM muscles as presented in Chin et al.’s [42] study, RF activity does track changes in workload to the same extent as other muscles do (i.e., VL, MG and LG) [41]. Indeed, Baum and Li [43] proved that lower extremity muscle activities during cycling are influenced by workload and also by frequency. In their study, a workload effect was observed from the onset of RF and the offset of RF muscles. Mileva et al. [44], reported that RF muscles demonstrated the largest fatigue index change across the 15 repetitions of bilateral knee extension exercise (80 ± 50% vs. 283 ± 89%; endurance vs. strength group). The average decrease in the median EMG frequency values for the strength and endurance groups was significantly different (−21 ± 5% and −9 ± 7%, respectively). A recent study conducted on power and endurance athletes showed that a significant difference (*p*  <  0.05) was found in the RF muscle of the dominant leg for several EMG features [45].

From a traditional point of view, muscle fatigue can be identified by a decrease in the frequency components of the EMG signal; for instance, detected from the MF of the SEMG power spectrum [46]. We know that there is an association between fiber type distribution and muscle fiber conduction velocity (MFCV) and/or spectral parameters such as MF and MPF of type II fibers [47]. Moreover, it has been shown that slowdown of MFCV leads to a decrease in MPF [48]. Therefore, it is likely that during cycling exercise the fatigue-induced shift to lower values of MPF are indicative of the differences in the MFCV characteristics (i.e., fiber type) of recruited muscle fibers [49]. In this study, during the progressive cycling exercise, we observed (in all groups) a shift towards lower EMG frequencies (i.e., a low-frequency fatigue) together with an increased AEMG. This relationship is considered to be an indicator of NMF [50]. Other studies have confirmed this mechanism, based on a higher AEMG signal toward the end of the exercise [51]. We observed that the rate of increase in AEMG during the submaximal cycling exercise was the highest in the CG group, whereas the STR group achieved significantly higher values of this parameter in relation to the END group. A significant reduction in MF in the END and CG groups (between 150 and 200 W) suggest that this SEMG parameter may be considered as a suitable indicator of muscle fatigue, which is in line with multiple previous studies [44,52,53]. Interestingly, such a change did not occur in the STR group.

In the present study, MPF decreased significantly at the peak workload (200 W) only in the CG group, while there was no change and even an increase between 150 and 200 W in both the STR group and the END group. These outcomes suggest decreased muscle activity in the CG group and no remarkable signs of NMF in either groups of athletes. When we focus on the groups of athletes, mean MF, but not the MPF variable, was significantly higher in the course of the cycling exercise in the END compared to the STR group (*p* = 0.038). However, the values of both were mostly similar in all groups during the last workload (200 W). Trends in these spectral parameters observed in the END and STR groups are not consistent with previous observations as higher values of the EMG power spectrum parameters (MF and MPF) have been observed for muscles with a greater percentage of type II fibers or greater relative area of type II fibers [54]. On this basis we should expect higher values of MF and MPF in strength-trained athletes, but these were higher in endurance-trained athletes (except MPF at 150 W). During submaximal, fatiguing exercise, these responses can be attributed to fatigue-induced increases in muscle activation and MU recruitment. It is probable that power and strength training improves MU recruitment and synchronization more than endurance training. Moreover, the greater NMF in the END group than in STR group could represent a specific defense mechanism to prevent any extensive peripheral fatigue as has been recently exposed in well-trained male cyclists [52].

RPE is considered a factor of subjective fatigue, also termed a psychological fatigue, and is highly correlated with physiological responses, muscle activity and workloads during both continuous incrementaland constant cycling [55,56]. Previous studies have shown a corresponding increase in the RPE and SEMG variables during cycling exercise using different protocols [11,57]. For example, the EMG and RPE responses have been shown to increase linearly until exhaustion during high-intensity cycling exercise, although a constant-load exercise protocol was used [58]. A similar observation was also present in this study when RPE and SEMG values gradually increased in accordance with the increase in exercise workload in each of the study groups. The results are clear, showing the highest RPE at exercise termination (200 W) in non-athletes (CG group), where peripheral fatigue was also the highest. However, significantly lower RPE was reported in the END group compared to other groups. If we refer to the previous studies of aerobic exercise anchored by RPE [59,60], we can conclude that the neuromuscular responses during submaximal exercises expressed by RPE were reflective of the fatigue-related physiological mechanisms. This mechanism corroborates our findings, and therefore RPE can be regarded as a useful indirect measure of muscle activity during progressive cycling exercise.

### Limitations

The main limitation of this study is the lack of normalizing SEMG measurement, which is the most appropriate for examining muscle activity during cycling for one-off measurements (i.e., sprint method). In the SEMG measurement we also did not include the upper limbs, which is a limitation because these muscle groups play a key role during balancing on the cycle ergometer. Furthermore, athletes from the END group were triathletes, so they routinely performed cycle exercise training. Hence, the type of exercise test was more specific for them, while for the athletes from the STR group it was the non-familiar mode. The study also involved young men, so the ability to generalize the results/conclusions to individuals of other genders and ages is limited. Lastly, it could be possible that the duration of the cycling test, according to our protocol (not to exhaustion), may not fit the time needed to obtain in-depth analysis for exercise-induced neuromuscular fatigue.

## 5. Conclusions

In conclusion, this study demonstrates that there is a significant variation in neuromuscular fatigue profiles between athletes with different training backgrounds when a cycling exercise is applied. The study confirmed a significant shift of the frequency spectrum of SEMG towards a lower frequency, together with the rising AEMG, when a participant is fatigued during progressive cycling exercise. Confirmation of this are the changes in MPF and MF between workloads of 150 and 200 W, especially in the END group, which suggest the onset of exercise-induced fatigue. The observed differences might be attributed to the fiber type specific characteristics of endurance- and strength-trained athletes. Probably due to the greater percentage of type II fibers and/or greater relative area of type II fibers in strength-trained athletes, they performed exercise cycling with better motor unit recruitment and synchronization compared to endurance-trained athletes. The effects comprised higher values of MF and MPF in endurance-trained athletes (except MPF at 150 W).

Furthermore, the approximately linear trends of the SEMG and RPE values of both groups of athletes (END and STR) with increasing workload support the increased skeletal muscle recruitment with perceived exertion or fatiguing effect. Thus, RPE may be useful to control intensity during progressive cycling exercise, providing an indirect measure of the muscle activity. However, greater increases in most SEMG parameters for the RF muscles of the quadriceps of endurance-trained athletes (triathlon) compared with strength-trained athletes (bodybuilding) suggest that the neuromuscular profile depends on the participant’s training background.

## Figures and Tables

**Figure 1 ijerph-19-08839-f001:**
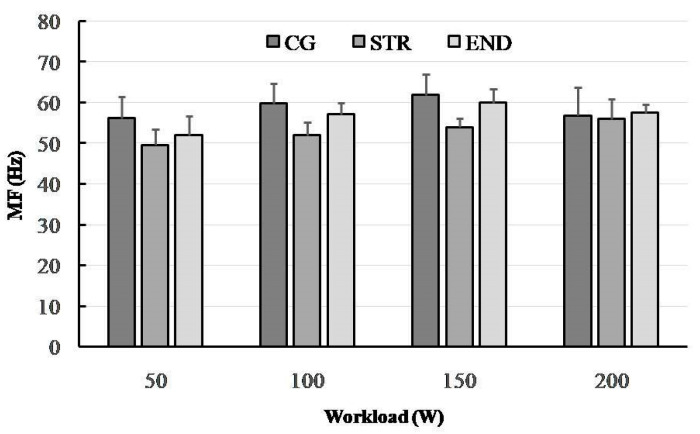
The changes in median frequency (MF) in the research groups (END—triathlon; STR—bodybuilder; CG—recreationally active) during progressive cycling exercise test at subsequent workloads; point and whiskers—mean ± SD.

**Figure 2 ijerph-19-08839-f002:**
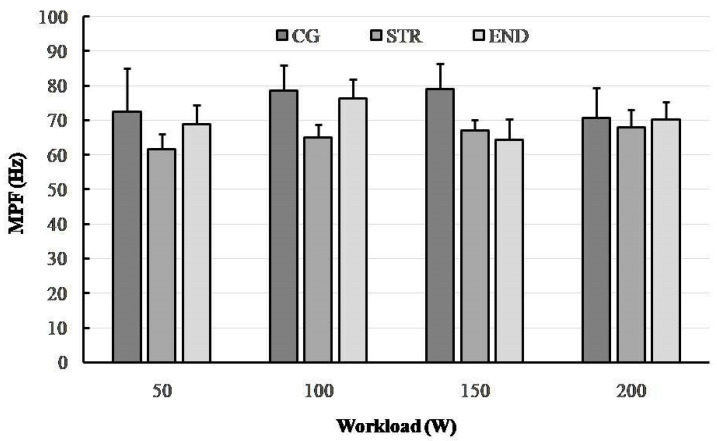
The changes in mean power frequency (MPF) in the research groups(END—triathlon; STR—bodybuilder; CG—recreationally active) during progressive cycling exercise test at subsequent workloads; point and whiskers—mean ± SD.

**Figure 3 ijerph-19-08839-f003:**
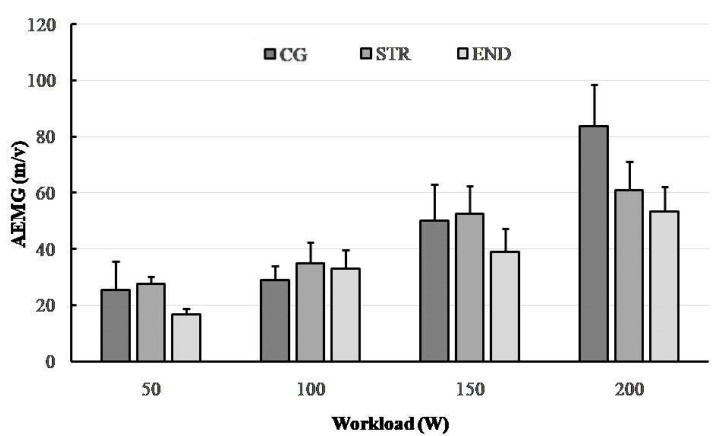
The changes in average amplitude of EMG (AEMG) in the research groups (END—triathlon; STR—bodybuilder; CG—recreationally active) during progressive cycling exercise test at subsequent workloads; point and whiskers—mean ± SD.

**Figure 4 ijerph-19-08839-f004:**
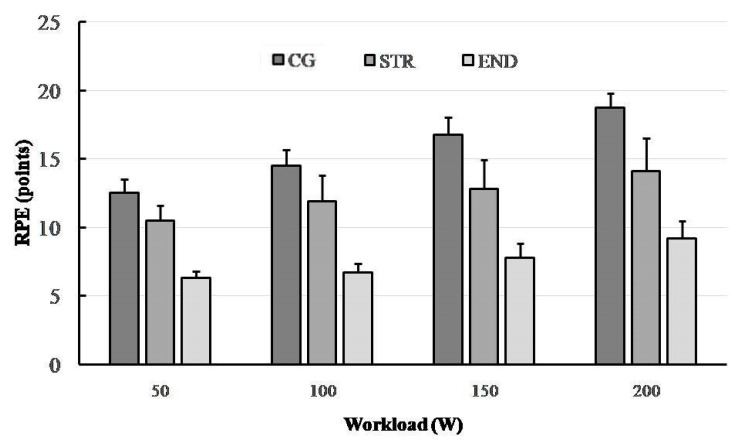
The changes in ratings of perceived exertion (RPE) in the research groups (END—triathlon; STR—bodybuilder; CG—recreationally active) during progressive cycling exercise test at subsequent workloads; point and whiskers—mean ± SD.

## Data Availability

Data are available upon request from the corresponding author.

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
