# Peer review of "Neuromuscular Fatigue Responses of Endurance- and Strength-Trained Athletes during Incremental Cycling Exercise"

_ijerph, 2022, doi:10.3390/ijerph19148839_

Round 1

Reviewer 1 Report

The obtained results can be extended to the use of physiological mechanisms of muscle fatigue development to correct the training process.

Author Response

We are grateful to the Reviewer for the time devoted to reviewing our manuscript and valuable comments.

Reviewer 2 Report

The authors examined the effect of exercise-induced neuromuscular and perceptual fatigue responses using significantly lower surface electromyography (SEMG) and the rating of perceived exertion (RPE) measurement during an incremental submaximal cycling exercise. The authors found that there was significant variation in neuromuscular fatigue profiles between athletes with different training backgrounds when a cycling exercise is applied. Endurance trained group had SEMG responses than those from strength trained and control group. Moreover, the linear trend of SEMG and rating perceived exertion values of endurance and strength trained group was observed.

There are several issues with this paper.

1.This article is not fully novel:

  • Garrandes, F., Colson, S. S., Pensini, M., Seynnes, O., & Legros, P. (2007). Neuromuscular fatigue profile in endurance-trained and power-trained athletes. Medicine and science in sports and exercise39 (1), 149–158. https://doi.org/10.1249/01.mss.0000240322.00782.c9

  • Hausswirth, C., Argentin, S., Bieuzen, F., Le Meur, Y., Couturier, A., & Brisswalter, J. (2010). Endurance and strength training effects on physiological and muscular parameters during prolonged cycling. Journal of electromyography and kinesiology : official journal of the International Society of Electrophysiological Kinesiology, 20(2), 330–339. https://doi.org/10.1016/j.jelekin.2009.04.008

  • Lepers, R., Maffiuletti, N. A., Rochette, L., Brugniaux, J., & Millet, G. Y. (2002). Neuromuscular fatigue during a long-duration cycling exercise. Journal of applied physiology (Bethesda, Md. : 1985), 92(4), 1487–1493. https://doi.org/10.1152/japplphysiol.00880.2001

  • Jurasz M, BoraczyÅ„ski M, Laskin JJ, Kamelska-Sadowska AM, Podstawski R, Jaszczur-Nowicki J, Nowakowski JJ, Gronek P. (2022). Acute Cardiorespiratory and Metabolic Responses to Incremental Cycling Exercise in Endurance- and Strength-Trained Athletes. Biology, 11(5):643. https://doi.org/10.3390/biology11050643

2. The title:

Line 2: The authors stated that this manuscript is a cross-sectional observational study (Methods section, line 91). It is recommended to include the type of the study in the title.

3. Abstract:

Line 21: Abbreviations are not described in the abstract (RPE scale, MF, MPF, HSD, AEMG).

Line 30: Highlighted part of the text has been observed. Highlighted words/letters have also occurred in the line 94, 100, 219, 233, 238, 240, 247, 248, 259, 260, 261, 264, 265. It is advised to remove the highlighting color from the text.

4. Introduction:

Line 40-43: The similarity in the sentence has occurred:

Garrandes et al. 2007: “Cross-sectional studies have reported that the training backgrounds of power-trained and endurance-trained athletes influence their respective neuromuscular profiles”.

Authors wrote: “Cross-sectional studies have reported that the training backgrounds of power- or strength-trained and endurance-trained athletes influence their respective neuromuscular fatigue (NMF) profiles”.

If the similarity was unintentional, the sentence should be placed in the quotation marks. Otherwise it could be considered as a plagiarism.

5. Materials and Methods:

Line 119-127: The similarity in 2.2. Eligibility criteria has occurred:

Jurasz M. et al. 2022: “Inclusion criteria for END and STR groups included: male, aged 18–30 years, self-reported regular engagement in endurance or strength training, at least 3 years of competitive experience, no current sustained experience in the other modes of training, nonsmokers, and no reported medical contraindications to participate in exercise testing. For the CON group, the inclusion criteria included: male, aged 18–30 years, non-smokers, no reported medical contraindications to participate in exercise testing, a BMI < 25 kg/m2 and active lifestyle (>600 MET) confirmed by an adapted version of short-form of International Physical Activity Questionnaire (IPAQ-SF)”.

Authors wrote: “The inclusion criteria for END and STR groups were: male, aged 18–30 years, self-reported regular engagement in endurance or strength training, at least 3 years of competitive experience, no current sustained experience in the other modes of training, non-smokers, not taking performance-enhancing drugs, and reporting no medical con-traindications to participate in exercise testing (e.g., knee injury or neuromuscular disorder). For the CON group the inclusion criteria included: male, aged 18–30 years, non-smokers, no reported medical contraindications to participate in exercise testing, a BMI < 25 kg/m2 and active lifestyle (> 600 MET) confirmed by an adapted version of short-form of International Physical Activity Questionnaire (IPAQ-SF)”.

The same situation happened in the 2.3. Ethical approval, 2.4. Design and procedures, 2.5. Pre-test preparation, 2.5. Anthropometry and 2.6. Incremental Cycling Test sections.

As I understand they are the same authors of both papers (Jurasz M, Boraczyński M, Laskin JJ, Kamelska-Sadowska AM, Podstawski R, Jaszczur-Nowicki J, Nowakowski JJ, Gronek P. Acute Cardiorespiratory and Metabolic Responses to Incremental Cycling Exercise in Endurance- and Strength-Trained Athletes. Biology. 2022; 11(5):643. https://doi.org/10.3390/biology11050643 and present manuscript) that collected the data from one study and analyzed different variables in two different papers. If such situation occurs, it is recommended to rewrite the Methods section, otherwise it is considered as autoplagiarism. On a positive note the authors rewrote correctly the 2.9. Statistical Analysis section.

6. Results:

Line 139: 2-week period is considered as a very short period for the observational studies. It is possible that the duration of the incremental cycling test and the lack of follow up could influence the results.

6. Discussion:

Line 406: Familiarization before the test could also influence the performance of the participants. Generally familiarization should enhance the results from the cycling test. However, it can also decrease the level of performance. It is recommended to divide familiarization and the main test in two different days to obtain reliable outcomes.

I do agree with the authors that there is a significant variation in neuromuscular fatigue profiles between different athletes during incremental cycling test. It merely depends on the duration of training (years), type of trained muscle group, different sports (bodybuilders versus triathletes) etc. This study had very active control group that obtained surprisingly the best results among all research groups (Figure 1, Figure 2, Figure 3, Figure 4). However, the paper is not fully consistent (Conclusion in the abstract is not the same as the 1 paragraph of Discussion and the last sentence of Discussion – please check the first paragraph of Discussion section), I would advise to correct it.

Author Response

(The authors gave the same response as above.)

Reviewer 3 Report

This study explored the development of neuromuscular fatigue responses during pro- 13 gressive cycling exercise.  The approximately linear trend of 27 the SEMG and RPE values of both groups of athletes with increasing workload support the in- 28 creased skeletal muscle recruitment with perceived exertion or fatiguing effect. Generally speaking, a native research method was presented and convinced results have been given.

Author Response

(The authors gave the same response as above.)

Reviewer 4 Report

The study of Jurasz et al. assesses the development of fatigue with an objective  (SEMG)  and subjective (RPE) method during progressive cycling exercise. The study is well designed and is an important contribution to the topic.

I have only minor comments:

1. Methods are very well written but please characterize briefly RPE Scale

2. Figures-please mark significant results with *

3. Discussion-are there any studies that compared SEMG between people performing different type of exercises or is the first study on this topic?

4. line 188 rightrectus-I think dominant (not right) is meant 

5. line 229- athletes(END)-space missing

6. Line 300-302 would it be possible to give some examples of these processes?

7. Line 312-317 and 346-347-the remark about muscle morphology is very important, however, maybe it would be better to put the part on muscle morphology together?

8. Conclusions-maybe it would be possible to shorten it a little to make them more clear e.g. I am not sure if the sentence "The SEMG signal is a valuable measure to study the lower-extremity muscle behavior under fatiguing cycling exercise, as it proves time-dependent changes" fits well to the Conclusion part

Author Response

(The authors gave the same response as above.)

Round 2

Reviewer 2 Report

The Authors improved the manuscript satisfactorily. I agree to accept it in present form.